# Synthetic Grafts in Anterior Cruciate Ligament Reconstruction Surgery in Professional Female Handball Players—A Viable Option?

**DOI:** 10.3390/diagnostics14171951

**Published:** 2024-09-03

**Authors:** Răzvan Marian Melinte, Dan Nicolae Zolog Schiopea, Daniel Oltean-Dan, Robert Bolcaș, Matei Florin Negruț, Tudor-Mihai Magdaș, Marian Andrei Melinte, Mircea Tăbăcar

**Affiliations:** 11st Department of Orthopaedics and Traumatology, “Iuliu Hațieganu” University of Medicine and Pharmacy, 47 Traian Mosoiu Street, 400394 Cluj-Napoca, Romania; 2Department of Orthopaedics and Traumatology, County Emergency Hospital, 47 Traian Mosoiu Street, 400394 Cluj-Napoca, Romania; 3MedLife Humanitas Hospital, 75 Frunzisului Street, 400664 Cluj-Napoca, Romania; 4Doctoral School of Medicine and Pharmacy, George Emil Palade University of Medicine, Pharmacy, Science and Technology, 1 Nicolae Iorga Street, 540142 Targu Mures, Romania; 5Department of Anaesthesia and Intensive Care II, “Iuliu Hațieganu” University of Medicine and Pharmacy, 3-5 Clinicilor Street, 400347 Cluj-Napoca, Romania; 6“Iuliu Hațieganu” University of Medicine and Pharmacy, 3-5 Clinicilor Street, 400347 Cluj-Napoca, Romania; 7Department of Anatomy, “Iuliu Hațieganu” University of Medicine and Pharmacy, 3-5 Clinicilor Street, 400006 Cluj-Napoca, Romania

**Keywords:** anterior cruciate ligament reconstruction (ACLR), autologous grafts, synthetic grafts, professional handball players, professional sports rehabilitation

## Abstract

Anterior cruciate ligament (ACR) rupture is a frequent injury in professional sports players. We conducted a retrospective cohort study, including 41 professional female handball players, undergoing ACR reconstruction surgery, using a Ligament Advanced Reinforcement System (LARS) graft (*n* = 12) or a Soft Tissue (ST) graft (*n* = 29). After return-to-play, the patients were asked to take a survey, reporting subjective and objective performance indexes before the injury and after return-to-play. Time from surgery to first practice and to return-to-play were significantly shorter in the LARS group (3.92 ± 1.14 vs. 6.93 ± 2.19 months, *p* < 0.001 and 4.71 ± 1.2 vs. 8.81 ± 2.9, respectively). While there was no difference between postoperative mean time on court, number of goals/match, number of matches played at 6 months return-to-play and 50 m, 100 m and gate-to-gate sprint times, there was a significantly greater increase in preoperative times in the ST group than in the LARS group (1.45 ± 1.05 s vs. 0.21 ± 0.58 s slower than preoperatively, *p* < 0.001 for 50 m; 1.09 ± 0.95 s vs. 0.08 ± 1 s, *p* = 0.01 for 100 m; 1.66 ± 1 s vs. 0.21 ± 0.66 s for gate-to-gate). In conclusion, LARS grafts provide a faster recovery time and better functional outcome, significantly impacting the performance of professional handball players.

## 1. Introduction

Anterior cruciate ligament (ACL) tears in professional athletes are one of the most frequent sports-related injuries [1]. Handball has a higher incidence of ACL lesions when compared to other team sports such as football, basketball and volleyball with women having a higher rate of ACL lesions than men [2,3]. In professional athletes, these injuries significantly impact the players’ careers, with only 69% returning to sports after a mean of 10.8 months, and often reporting lower performance levels compared to those previous to the injury. However, when considering elite athletes, this percentage increases to 79% [4].

There are currently no clear guidelines regarding graft choice in anterior cruciate ligament reconstruction (ACLR) surgery [5]. Moreover, no consensus has been reached regarding the role and indications of extra-articular reinforcement techniques [6], the impact of surgical techniques (such as tunnel placement, graft selection and graft fixation) on the re-rupture rate, the postoperative rehabilitation process [7,8], and rigorous return to play criteria based on isokinetic tests, hop tests, stability measurement, and MRI findings [9].

Regarding graft selection, there is a wide variety of options including auto-, allo-, and synthetic grafts [5,10]. Although autografts are widely used and with promising results, the main disadvantages include donor-site morbidity (the patellar tendon and hamstring muscles all have specific roles in knee biomechanics) and a prolonged rehabilitation process needed for the integration of the biological graft. In contrast, newer synthetic grafts are emerging as a promising alternative to autografts, bringing along a series of theoretical advantages, including no donor site morbidity, a faster recovery and prompter return to play, with the disadvantage of a slightly higher re-rupture rate [11]. Compared to allografts, the synthetic grafts have a lower risk of failure in the early postoperative phase, thus allowing for an accelerated recovery protocol, reducing the rehabilitation time from an average of 10–11 months to 4–6 months [4,12].

The Ligament Advanced Reinforcement System (LARS) is composed of a synthetic polyethylene terephthalate (PET) structure, which is designed to mimic the mechanical properties of the native ACL. PET has high tensile strength and stiffness, making it a suitable material for synthetic grafts. Its intraosseous segment is composed of longitudinal fibers bound together by a transverse knitted structure, while the intra-articular segment consists of multiple parallel longitudinal fibers twisted at a 90-degree angle. Biomechanical studies have demonstrated that LARS provides comparable load-bearing capacity and stiffness to traditional autografts and allografts. LARS has been shown in vitro to have excellent fatigue resistance, ensuring long-term durability under repetitive loading conditions [13,14].

The aim of this study was to compare the short- and medium-term outcomes of anterior cruciate ligament reconstruction (ACLR) surgery, using synthetic third-generation LARS systems or autologous hamstring grafts, in professional female handball players, through subjective and objective parameters, while assessing knee function and specific performance measurements.

## 2. Materials and Methods

### 2.1. Patient Selection

We conducted a single-center, retrospective cohort study, including professional female handball players who sustained a non-contact sports injury resulting in an ACL rupture. We included consecutive patients, operated between February 2019 and April 2022, in the Department of Orthopaedics and Traumatology of MedLife Humanitas Hospital, Cluj-Napoca, Romania, using either a synthetic third-generation LARS system or autologous hamstring grafts. The diagnosis was established following history, clinical examination, knee radiographs, MRIs, and confirmed arthroscopically at the time of definitive surgery. Exclusion criteria were multiple ligament injuries, unstable meniscal tears, bony deformities (increased posterior tibial slope, genu varum/valgus of more than 5 degrees, ipsilateral previous knee surgery), or revision ACL surgery. Patients with partial meniscus resections and mild cartilage lesions were included. All patients agreed to take part in the study, and written informed consent was obtained prior to inclusion.

### 2.2. Preoperative Management

All patients underwent at least two weeks of prehabilitation. The surgery was performed by the same senior surgeon and surgical team carrying out more than 150 ACL reconstructions/year.

The patients were informed about the advantages and disadvantages of each graft type and chose according to their preferences. The patients who opted for a synthetic graft were included in the LARS group, while those opting for an autograft were included in the Soft Tissue (ST) group.

### 2.3. Surgical Management

Patients in the LARS group had an arthroscopic ACLR with a 4 mm double-folded LARS (3rd Generation Ligament Augmentation Reconstruction System—product code 104.106, Corin Group, Cirencester, UK), through an adjustable ACL TightRope II, resulting in an 8 mm diameter synthetic graft. The femoral and tibial tunnels were drilled in a remnant-preserving way to decrease the mechanical stress of the graft at the tunnel entrance. Femoral fixation was enhanced using an 8 mm titanium interference screw and an extracortical button. Tibial fixation was achieved using an 8 mm titanium interference screw and a titanium anchor. Tunnels were in an isometric non-anatomical position, and no extra-articular reinforcement procedures were added.

For the patients in the ST group, arthroscopic ACLR was carried out, using a quadrupled semitendinosus and gracilis graft with an isometrical tunnel placement. All grafts below 7 mm in diameter were braided to increase the diameter and tensile strength, while grafts between 7 and 8 mm were reinforced using FiberTape, which is an ultra-high-strength non-absorbable suture (Arthrex, Naples, FL, USA). Femoral cortical fixation was achieved using an adjustable loop button, ACL TightRope II (Arthrex, Naples, FL, USA), while bioresorbable interference screws were utilized for tibial fixation of the graft (Arthrex, Naples, FL, USA).

### 2.4. Postoperative Rehabilitation and Return to Play

All patients were allowed to weight bear as tolerated immediately after surgery, and the postoperative rehabilitation protocol commenced the first day after surgery. The rehabilitation program was focused on achieving strength, normal range of motion, dynamic knee stability and was based on practice guidelines provided by van Melick et al. [15]. Rehabilitation lasted for 4–8 months. Rehabilitation duration was influenced by patients’ compliance and adherence to the rehabilitation protocol, individual pain toleration and psychological readiness for return to play.

Return to play was permitted after passing several dynamic stability tests (LESS score, single-leg hop test, triple single-leg hop test) and after achieving a minimum of 95% on the Limb Symmetry Index while measuring isokinetic muscle strength on an Easy Torque machine (TONUS Sports & Reha GmbH, Zemmer, Germany). Aerobic endurance and psychological readiness were also included in the return-to-play criteria. All patients had the maximum of 100 points in the International Knee Documentation Committee (IKDC) and Lysholm scores before returning to play. We did not test instability of knee on specific devices.

### 2.5. Data Collection

Two years after the surgery, all patients were required to fill a survey in which they were asked to provide some objective and subjective parameters (Appendix A). From this survey, data were collected regarding patients’ age at the time of the surgery, graft type and pre- and postoperative average time spent on court during a game, number of goals scored per game, gate to gate, 50 m and 100 m sprint times, number of anti-inflammatory tablets administered per week and number of matches played in the 6 months prior to injury and 6 months following return-to-play. Data regarding whether the patients played in the national team before or after surgery were also collected, as well as time from the surgery to the first practice, time from surgery to first game and postoperative subjective gameplay performance at 12 months (graded by the patients on a scale from 1 to 10, 10 being equal to their performance before the surgery). Re-rupture rates were also recorded.

### 2.6. Statistical Analysis

Statistical analysis was performed using SPSS Statistics v29.0.1 (IBM Corp., Armonk, NY, USA). Continuous variables were reported as mean ± standard deviation (SD), and categorical variables were reported as frequencies. The normality of continuous variables was tested using Shapiro–Wilk’s test, and equality of variances was tested using Levene’s test. Data following a normal distribution were compared using Student’s *t*-test, and those not normally distributed were compared using a Mann–Whitney U-test. When analyzing data from paired samples, the t-test for paired samples or a Wilcoxon rank-sum test were used. Categorical variables were compared using a Chi-square square test or Fisher’s exact test. Effect size was evaluated using Hedges’ g, and correlation analysis was performed using point-biserial correlation coefficient (r_pb_). A two-tailed *p* < 0.05 was considered statistically significant.

## 3. Results

Forty-eight female handball players undergoing ACLR were screened for inclusion. Of those, one patient had previous ipsilateral knee surgery, one was diagnosed with multiple ligament injury, two were excluded for unstable meniscal tears, two were excluded for bone deformity—increased tibial slope >10 degrees and genu varum >5 degrees—and one failed to complete all relevant questions in the survey, leaving 41 patients to be included in the final analysis. Some patients underwent meniscus resections, but none had meniscus suture, while several patients had mild cartilage lesions (Outerbridge I/II). Twelve patients were included in the LARS group, and 29 patients were included in the ST group. The baseline and postoperative characteristics of all patients and the two subgroups are shown in Table 1.

Patient age ranged from 18 to 34. No statistically significant difference was found between the age of patients in the two groups, the mean time spent on court per match preoperatively, mean number of goals per match, or number of matches played in the 6 months prior to injury. However, more of the patients in the ST group played in the national team before surgery (79.31% versus 41.67%, *p* = 0.029), and they had a shorter preoperative gate-to-gate sprint time (7.33 ± 1.18 vs. 8.33 ± 1.37 s, *p* = 0.014) and 50 m sprint time (8.17 ± 1.4 vs. 9.04 ± 1.42, *p* = 0.011) than the LARS group. The preoperative 100 m sprint time and number of anti-inflammatory tablets administered per week were not significantly different between the groups.

Postoperative parameters did not differ significantly between the two groups (postoperative mean time on court per match, mean number of goals per match, number of matches played in the first 6 months after return to field, percent of players joining the national team, gate to gate, 50 m and 100 m sprint times and anti-inflammatory medication administration).

A comparison of preoperative and postoperative parameters is provided in Table 2. For the ST group, a statistically significant decrease was found between pre- and postoperative mean time on court/match (49.91 ± 12.56 vs. 33.71 ± 16.42 min, *p* < 0.001) and the number of matches played 6 months prior to injury versus 6 months post-return to field (24.33 ± 12.64 vs. 17.29 ± 10.95, *p* = 0.004). The gate-to-gate sprint time (7.33 ± 1.18 vs. 8.98 ± 1 s, *p* < 0.001), 50 m sprint time (8.17 ± 1.4 vs. 9.62 ± 1.57 s, *p* < 0.001) and 100 m sprint time (14.19 ± 1.12 vs. 15.28 ± 1.17 s, *p* < 0.001) all significantly increased postoperatively in the ST group. The mean number of goals scored per match and the number of anti-inflammatory tablets administered per week did not differ between the pre-injury and postoperative period.

In the LARS group, none of the parameters mentioned previously differed between the baseline and the postoperative period except for the number of matches played in a 6-month interval (24.25 ± 14.65 pre-injury vs. 14.04 ± 13.18 postoperatively, *p* = 0.012).

Table 3 summarizes the main postoperative outcomes and changes in performance parameters. Patients in the LARS group had a significantly shorter return-to-practice time (3.92 ± 1.14 vs. 6.93 ± 2.19 months, *p* < 0.001) and return-to-game time (4.71 ± 1.2 vs. 8.81 ± 2.9 months, *p* < 0.001) than those in the ST group. This accounts for a large effect size (Hedges’ g = 1.56 for return-to-practice and g = 1.64 for return-to-game) and a moderate correlation between LARS and shorter return-to-practice time (r_pb_ = 0.58, *p* < 0.01) and shorter return-to-game time (r_pb_ = 0.6, *p* < 0.001), respectively.

Moreover, those in the ST group showed a significantly greater increase from baseline in their gate-to-gate sprint time when compared to the LARS group, both as absolute values (1.66 ± 1 vs. 0.21 ± 0.66 s, *p* < 0.001) and as percentages of their pre-injury times (24.41 ± 17.04 vs. 2.13 ± 7.05%, *p* < 0.001). This accounts for a large effect size (Hedges’ g = 1.66 for absolute increase and g = 1.58 for percentual increase) and a moderate correlation between ST and larger increase in gate-to-gate sprint time (r_pb_ = 0.59, *p* < 0.001 for absolute increase and r_pb_ = 0.57, *p* < 0.001 for percentual increase).

The postoperative increase (compared to pre-injury times) in 50 m sprint time (1.45 ± 1.05 vs. 0.21 ± 0.58, *p* < 0.001; 18.59 ± 14.04 vs. 2.39 ± 6.69%, *p* < 0.001) and 100 m sprint time (1.09 ± 0.95 vs. 0.08 ± 1, *p* = 0.01; 7.9 ± 7.19 vs. 0.71 ± 6.94%, *p* = 0.012) was also significantly greater in the ST group compared to the LARS group. The effect size was large for all parameters (Hedges’ g = 1.66 for absolute increase in 50 m sprint time; g = 1.30 for percentual increase in 50 m sprint time; g = 1.04 for absolute increase in 100 m sprint time; g = 1.01 for percentual increase in 100 m sprint time). There was a moderate correlation between ST and greater increase in 50 m sprint time (r_pb_ = 0.52, *p* < 0.001 for both absolute increase and percentual increase) and a weak correlation between ST and greater increase in 100 m sprint time (r_pb_ = 0.44, *p* = 0.004 for absolute increase and r_pb_ = 0.43, *p* = 0.005 for percentual increase).

Postoperative subjective gameplay performance scores did not differ between the groups. When comparing pre-injury to post-return-to-game performance parameters, the decrease in the number of goals scored per match, the difference in the number of matches played in a 6-month period and the difference in mean time on court per match did not differ significantly between the two groups.

No ligament re-rupture was reported after a 2-year follow-up.

## 4. Discussion

Our study aimed to provide an answer regarding the graft selection process for ACLR in professional athletes and whether synthetic grafts can represent an adequate alternative to the traditional soft tissue grafts.

The results of our study showed that the use of LARS provided more favorable outcomes regarding time elapsed from surgery to return-to-practice and return-to-play, a better sprinting profile (assessed by measuring gate-to gate, 50 m and 100 m sprint times) and no differences regarding subjective gameplay performance, numbers of matches played, goals-per-game, or minutes-per game.

A shorter recovery period represents, in our opinion, one of the most significant desiderates while facing ACLR in the elite athlete, and synthetic grafts bring this advantage, allowing the athletes to take part in important sporting events.

No other studies are available for comparison to date, investigating the effectiveness and safety profile of LARS versus ST grafts in ACLR in this specific patient population. In a recent study by Jones et. al. [16], the LARS system was shown to be safe and effective in professional athletes undergoing extra-articular knee ligament reconstructions (medial collateral ligament and posterolateral corner reconstruction), allowing for 88.2% of athletes to return to play. However, the study population differed significantly from our patients, as Jones et. al. included athletes playing multiple sports, mostly football and rugby players, as opposed to our cohort of female handball players.

A 2020 meta-analysis by Sun et al. [17] confirmed the superiority of the synthetic grafts over the autografts, providing better knee joint stability, patient-reported outcome scores and fewer postoperative complications with no difference in graft failure in any of the included studies.

First and second-generation synthetic grafts were associated with an increased rate of complications including graft failures, knee instability, extensile synovitis, and early onset arthritis, which led to the abandonment of these grafts. Third-generation LARS provide an improved design which decreases the postoperative complications with excellent short and mid-term results [12]. However, the long-term results of LARS-ACLRs remained controversial, with some studies showing an increased graft failure rate [18], while others conclude that when using a remnant preserving technique, the synthetic graft showed satisfactory outcomes at 10 years postoperatively when compared to a soft tissue graft [19].

A 2020 histological analysis of failed ACL-LARS reconstructions [20] showed a poor bone and soft-tissue ingrowth of the synthetic graft and was cited along mechanical issues (poorly positioned tunnels) as potential factors for the early failures. The most frequent location of failure is described as being at the femoral bone tunnel aperture [21], the main mechanism being yarn-on-bone wear [22]. Consequently, we recommend that all efforts should be directed to an optimal femoral tunnel placement, in an isometric, anatomic position, while the shape of the aperture should be oval-shaped to further decrease the stress on the graft. This improved mechanical environment could also address another significant clinical concern, represented by the presence of foreign-body synovitis, which is described in the literature [23].

Tissue engineered constructs also represent an exciting prospect and can provide in the future a solution regarding the improvement of the biology of the synthetic grafts [24].

Therefore, it is our opinion that in order to decrease the incidence of these potential issues, ACLR with LARS should only be used in select cases: the procedure should only be performed by high-volume surgeons, specialized in sports traumatology, in a stump preserving manner, to preserve the local biology and achieve optimal-placed bone tunnels.

An additional factor to be considered is that the long-term results in ACLR in female athletes are poorer compared to ACLR in non-professional athletes with higher re-rupture rates with all graft choices [25].

Additional extra-articular procedures should be added based on recent findings and can be performed regardless of graft choice especially in the female athlete population.

However, it is crucial to proceed with caution, recognizing the need for further research into long-term efficacy and potential complications. Optimized surgical techniques and advances in tissue engineering may further enhance the utility and success of synthetic grafts in ACL reconstruction, suggesting their selective use in professional athletics, facilitated by experienced surgeons to ensure optimal outcomes. Correct patient information and shared decision making should be mandatory in these cases.

Our study has several limitations, including a small sample size and non-randomized design. The number of patients differs between the two groups (12 in the LARS group and 29 in the ST group) and the patients were able to choose the type of graft based on personal preference, which might influence their subjective perception of postoperative performance. However, to the best of our knowledge, no other studies involving a larger patient population are available to date, focusing solely on professional female handball players.

## 5. Conclusions

Our study conclusively demonstrates that synthetic grafts, specifically the Ligament Advanced Reinforcement System (LARS), offer a viable and effective alternative to autografts for ACL reconstruction in professional female handball players, notably reducing recovery times and enabling quicker return to sport without compromising performance.

## Figures and Tables

**Table 1 diagnostics-14-01951-t001:** Preoperative and postoperative patient characteristics.

Demographics	All Patients (*n* = 41)	LARS (*n* = 12)	ST (*n* = 29)	Test Parameter	Effect Size	Statistical Significance
Age	25.66 ± 4.46	24.17 ± 3.83	26.28 ± 4.61	1.39 *	0.48 *	0.171
Preoperative mean time on court/game (min)	48.05 ± 13.82	43.54 ± 16.18	49.91 ± 12.56	143.5 **	0.14 **	0.392
Preoperative mean number of goals/game	5.24 ± 2.76	5.75 ± 2.83	5.07 ± 2.77	133.5 **	0.06 **	0.725
Number of matches played in the 6 months prior to injury	24.3 ± 13.07	24.25 ± 14.65	24.33 ± 12.64	0.02 *	0.01 *	0.986
Played on the national team before surgery				4.78 ***	0.34 ***	0.029
Yes, *n* (%)	28 (68.29%)	5 (41.67%)	23 (79.31%)			
No, *n* (%)	13 (31.71%)	7 (58.33%)	6 (20.69%)			
Preoperative gate to gate sprint time (s)	7.62 ± 1.31	8.33 ± 1.37	7.33 ± 1.18	87.5 **	0.4 **	0.014
Preoperative 50 m sprint time (s)	8.43 ± 1.44	9.04 ± 1.42	8.17 ± 1.4	84 **	0.42 **	0.011
Preoperative 100 m sprint time (s)	14.45 ± 1.44	15.08 ± 1.93	14.19 ± 1.12	127 **	0.22 **	0.185
Preoperative number of anti-inflammatory tablets/week	1.21 ± 2.54	0.83 ± 1.34	1.36 ± 2.9	168 **	0.03 **	0.875
Postoperative mean time on court/game (min)	34.49 ± 17.68	36.38 ± 21.09	33.71 ± 16.42	0.44 *	0.15 *	0.666
Postoperative mean number of goals/game	4.76 ± 3.14	5.25 ± 3.64	4.59 ± 3	0.56 *	0.19 *	0.517
Numbers of matches played in the first 6 months after return to the field	16.34 ± 11.58	14.04 ± 13.18	17.29 ± 10.95	138 **	0.16 **	0.311
Played on the national team after surgery				0.77 ***	0.14 ***	0.38
Yes, *n* (%)	18 (43.90)	4 (33.33%)	14 (48.28%)			
No, *n* (%)	23 (56.10)	8 (66.67%)	15 (51.72%)			
Postoperative gate to gate sprint time (s)	8.85 ± 1.25	8.54 ± 1.72	8.98 ± 1	140 **	0.16 **	0.339
Postoperative 50 m sprint time (s)	9.51 ± 1.55	9.25 ± 1.54	9.62 ± 1.57	165.5 **	0.04 **	0.82
Postoperative 100 m sprint time (s)	15.24 ± 1.43	15.17 ± 1.99	15.28 ± 1.17	0.18 *	0.06 *	0.862
Postoperative number of anti-inflammatory tablets/week	1.84 ± 2.63	1.83 ± 3.16	1.84 ± 2.44	160.5 **	0.06 **	0.711

* For normally distributed variables, test parameter is reported as Student’s t and effect size is reported as Cohen’s d. ** For variables not following the normal distribution, test parameter is reported as Mann–Whitney U and effect size is reported as r. *** For dichotomous variables, test parameter is reported as Chi square and effect size is reported Cramer’s V.

**Table 2 diagnostics-14-01951-t002:** Comparison of baseline and postoperative parameters in the patients’ groups.

	LARS (*n* = 12)			ST (*n* = 29)		
	Preoperative	Postoperative	*p*	Preoperative	Postoperative	*p*
Mean time on court/game (min)	43.54 ± 16.18	36.38 ± 21.09	0.332	49.91 ± 12.56	33.71 ± 16.42	<0.001
Mean number of goals/game	5.75± 2.83	5.25 ± 3.64	0.472	5.07 ± 2.77	4.59 ± 3	0.222
Number of matches played in the 6 months prior to injury	24.25 ± 14.65	14.04 ± 13.18	0.012	24.33 ± 12.64	17.29 ± 10.95	0.004
Gate to gate sprint time (s)	8.33 ± 1.37	8.54 ± 1.72	0.295	7.33 ± 1.18	8.98 ± 1	<0.001
50 m sprint time (s)	9.04 ± 1.42	9.25 ± 1.54	0.241	8.17 ± 1.4	9.62 ± 1.57	<0.001
100 m sprint time (s)	15.08 ± 1.93	15.17 ± 1.99	0.777	14.19 ± 1.12	15.28 ± 1.17	<0.001
Number of anti-inflammatory tablets/week	0.83 ± 1.34	1.83 ± 3.16	0.326	1.36 ± 2.9	1.84 ± 2.44	0.183

**Table 3 diagnostics-14-01951-t003:** Postoperative outcomes in the two patients’ group.

Demographics	All Patients (*n* = 41)	LARS (*n* = 12)	ST (*n* = 29)	Test Parameter	Effect Size	Statistical Significance
Months from surgery to first practice	6.05 ± 2.37	3.92 ± 1.14	6.93 ± 2.19	5.76 *	1.98 *	<0.001
Months from surgery to first game	7.61 ± 3.14	4.71 ± 1.2	8.81 ± 2.9	29 **	0.65 **	<0.001
Postoperative subjective gameplay performance	8.37 ± 1.28	8.46 ± 1.03	8.33 ± 1.39	0.29 *	0.11 *	0.771
Increase in gate-to-gate sprint time (s)	1.23 ± 1.12	0.21 ± 0.66	1.66 ± 1	41 **	0.61 **	<0.001
Increase in gate-to-gate sprint time (% from baseline)	17.89 ± 17.95	2.13 ± 7.05	24.41 ± 17.04	31 **	0.65 **	<0.001
Increase in 50 m sprint time (s)	1.09 ± 1.09	0.21 ± 0.58	1.45 ± 1.05	55.5 **	0.55 **	<0.001
Increase in 50 m sprint time (% from baseline)	13.85 ± 14.35	2.39 ± 6.69	18.59 ± 14.04	51 **	0.56 **	<0.001
Increase in 100 m sprint time (s)	0.79 ± 1.05	0.08 ± 1	1.09 ± 0.95	83.5 **	0.42 **	0.01
Increase in 100 m sprint time (% from baseline)	5.8 ± 7.77	0.71 ± 6.94	7.9 ± 7.19	85 **	0.41 **	0.012
Decrease in mean number of goals/game	0.49 ± 2.05	0.5 ± 2.64	0.48 ± 1.87	173 **	0.02 **	0.982
Difference in the number of matches played 6 months post-return to field compared to 6 months prior to injury	7.96 ± 12.04	10.21 ± 11.79	7.03 ± 12.22	146.5 **	0.12 **	0.43
Difference in mean time on court/game (min)	13.2 ± 19.1	7.17 ± 25.74	15.79 ± 15.29	130 **	0.2 **	0.195

* For normally distributed variables, test parameter is reported as Student’s t and effect size is reported as Cohen’s d. ** For variables not following the normal distribution, test parameter is reported as Mann–Whitney U and effect size is reported as r.

## Data Availability

Data are contained within the article.

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
