# Peer review of "Synthetic Grafts in Anterior Cruciate Ligament Reconstruction Surgery in Professional Female Handball Players—A Viable Option?"

_diagnostics, 2024, doi:10.3390/diagnostics14171951_

Round 1

Reviewer 1 Report

Comments and Suggestions for Authors

Dear,

please find my comments attached.

Kind regards

Author Response

Thank you for your comprehensive and prompt review. We modified the manuscript according to the reviewer's requests, as follows:

  1. We have added an explanation of the factors that influenced the patient's rehabilitation duration, in the Materials and Methods section, Line 130.
  2. We have referred the mentioned survey in the main text. We believe that the survey we conducted is reliable, as professional athletes track their performance parameters closely, and most of our patients recorded sprint times, number of goals and time played per match systematically, both before their injury and after return-to-play.
  3. We have added the survey as "Appendix A".

Reviewer 2 Report

Comments and Suggestions for Authors

General Comments

The aim of this study was to compare the short- and medium- term outcomes of anterior cruciate ligament reconstruction (ACLR) surgery, using synthetic third generation LARS systems or autologous hamstring grafts, in professional female handball players. It is an interesting manuscript a quite methodological procedures and actual research topic, however some sections need improvements. Please, consider the following point-by-point revisions in the specific comments.

Specific Comments

Introduction: This section should be expanded by adding some previously published evidence on this research topic.

Materials and methods: Please restructure the materials and methods section according to the STROBE statement (it should report the sections on participants, research design, selection criteria, procedures, variables, statistical analysis, etc.). Some critical points are not described, such as: (1) basic characteristics and representativeness of the sample; (2) ethical procedures; (3) study design and selection criteria; (4) study variables, taking into account the procedures described. All variables described in the results section should be described in the results.

Results: Please add the remaining inferences to the presentation of the results in tables 1 to 3. Also, the authors should restate the effect size in the chi square taking into account Crammer V. The authors only report the p-value, but should include the t/w, mean differences and effect size.

Discussion: The discussion should endeavor to compare the correlation values between variables found in this study in relation to previous studies. The same goes for the regression coefficients. Please focus on the discussion of your results and not on theoretical assumptions that your study only allows to speculate on. Also, delve into the practical applications, future prospects and the study's limitations.

Conclusions: What inference did you use to formulate this statement? Please clarify.

Author Response

We are grateful for the review, and we believed that it has brought some valuable improvements to our paper. We have made the following alterations to the manuscript, as requested by the reviewer:

  1. Introduction - we expanded this section, by adding a few sentences on the advantages and disadvantages the two different graft types. As no previous articles comparing LARS and ST grafts in this specific patient population have been previously published, we could not cite specific data regarding professional female handball players.
  2. Materials and Methods - we restructured this section, providing subheadings, and added information regarding the ethical procedure, study design and study variables. The representativeness of the sample could not be assessed, as no previous data has been published on the topic and no objective parameters have been standardized for reporting the outcome in this specific patient population, so the minimum sample size could not be calculated. We instead included all consecutive patients operated in our clinic, that did not have any of the exclusion criteria.
  3. Results - we added the required parameters and updated Tables 1 and 3.
  4. Discussion - we expanded this section by comparing our data to some previously published studies.
  5. Conclusion - We find our data supports this statement, as both the interval between the surgery and first practice and from surgery to first game were significantly shorted in the LARS groups than the ST group. Moreover, postoperative sprint time increased less when compared to preoperative sprint times in the LARS groups than in the ST group. As no ligament re-tears have been reported in the study population, and based on our experience with both types of grafts, we are confident to say that the LARS system provides a safe and effective alternative to autografts for ACL reconstruction.

We are looking forward to hearing from you soon and hope you will consider our manuscript for publication.

King regards.

Round 2

Reviewer 2 Report

Comments and Suggestions for Authors

Dear author, 

After the extensive and careful revision in the first round, I recommend accepting the article in its present form. 

Best regards, 

José Eduardo Teixeira.